TOPICAL REVIEW

# Human models for COVID-19 research

Maximillian N. J. Woodall, Tereza Masonou 🆔, Katie-Marie Case and Claire M. Smith 🆔

*GOS Institute of Child Health, University College London, London, UK*

Edited by: Ian Forsythe & Frank Powell

The peer review history is available in the Supporting Information section of this article (https://10.1113/JP281499#support-information-section).

**Abstract** Currently, therapeutics for COVID-19 are limited. To overcome this, it is important that we use physiologically relevant models to reproduce the pathology of infection and evaluate the efficacy of antiviral drugs. Models of airway infection, including the use of a human infection challenge model or well-defined, disease relevant *in vitro* systems can help determine the key

**Claire M. Smith**, PhD, is an Associate Professor in the department of Infection, Immunity and Inflammation, GOS Institute of Child Health, University College London. Her research focuses on anti-viral therapies, mucosal immunity and the interaction between the airway epithelium and neutrophils during respiratory virus infection. **Maximillian Woodall**, PhD, is a Research Fellow working alongside C.M.S. His research employs an experimental model of the human airway to investigate age-dependent factors driving COVID-19 disease severity. **Tereza Masonou**, MSc, is a PhD student working under the supervision of C.M.S. Her research investigates neutrophil transepithelial migration and function during SARS-CoV2 infection, and how this differs between older adults and children. **Katie M. Case**, MSc, is a Research Assistant working alongside M.W. She maintains airway epithelial cells from different age groups and explores baseline differences such as cell motility, cilia beating and protein levels.

components that perpetuate the severity of the disease. Here, we briefly review the human models that are currently being used in COVID-19 research and drug development.

(Received 23 April 2021; accepted after revision 12 July 2021; first published online 19 July 2021)

**Corresponding author** C. M. Smith: UCL GOS Institute of Child Health, 30 Guilford Street, London, UK.    Email: c.m.smith@ucl.ac.uk

**Abstract figure legend** Current physiologically relevant models for studying COVID19. Image made using BioRender.com.

## Introduction

Coronavirus disease 2019 (COVID-19)is an infectious disease caused by a newly emerged severe acute respiratory syndrome (SARS)-coronavirus-2 (SARS-CoV-2). COVID-19 presents a diverse clinical spectrum, ranging from an asymptomatic carrier state to patients with life-threatening multi-organ failure and death (Huang *et al.* 2020*a*). The greatest risk factor for severe disease is age, with higher morbidity and mortality rates in the elderly population, despite younger people shedding similar levels of virus (Yanez *et al.* 2020). The overall case fatality rate of COVID-19 is 2.3%, rising to 14.8% in patients over the age of 80, and 49% among the critically ill (Kang & Jung, 2020).

The human airway epithelium is the first line of defence against inhaled pathogens and, to help prevent damage to the vulnerable tissues, the airways have developed physical and innate molecular barriers. These include polyps, hairs and mucus to trap foreign objects. At the microscopic level, the airway epithelial barrier is composed of differentiated cell types including muco-secretory goblet cells and ciliated cells (Pack *et al.* 1981). These cells function together to maintain a healthy homeostasis through the production of secretions that regulate the volume and viscosity of the fluid layer, and motile ciliary beating that coordinates clearance (Empey & Kolls, 2017).

SARS-CoV-2 virus enters the airway through the oral and nasal cavities (Gallo *et al.* 2020). The inhaled virus can evade these initial barriers and infect the respiratory epithelium. SARS-CoV-2 is thought to primarily target and infect airway epithelial cells via interaction of the viral spike glycoprotein with the angiotensin converting enzyme 2 (ACE2) host cell receptor (Zhou *et al.* 2020*b*; Hoffmann *et al.* 2020), although other receptors have been implicated (i.e. NRP1, BSG, TFRC) (Wang *et al.* 2020; Tang *et al.* 2020; Cantuti-Castelvetri *et al.* 2020; Daly *et al.* 2020). An impressive body of RNA sequencing data has confirmed that *ACE2* is expressed in approximately 1% of epithelial cells (Ziegler *et al.* 2020) and there is a gradient of ACE2 expression from the upper to the lower airways (Hou *et al.* 2020). Within the airways, ACE2 protein is thought to be most abundantly found on ciliated cells, nasal goblet cells and alveolar type II cells (Sungnak *et al.* 2020; Li *et al.* 2020; Ortiz *et al.* 2020; Lee *et al.* 2020*b*). There is also evidence that some secretory cell types, notably the 'secretory3' cell type (an intermediate position between club or goblet cells and ciliated cells) may be an important viral target as they are ACE2$^+$ and co-express viral spike priming proteases transmembrane serine protease 2 (TMPRSS2) and/or furin (Sungnak *et al.* 2020; Lukassen *et al.* 2020; Ziegler *et al.* 2020; Hou *et al.* 2020; Schuler *et al.* 2021). Histological examination of the lungs of deceased COVID-19 patients showed it was predominantly the ciliated cells that were infected, whereas MUC5B$^+$ club cells, MUC5AC$^+$ goblet cells and p63$^+$ basal cells were not infected (Hou *et al.* 2020; Schaefer *et al.* 2020). Alveolar type 1 (AT1) and 2 (AT2) cells (or AT2 cells that had transitioned to AT1 cells) can also become infected (Hou *et al.* 2020; Schaefer *et al.* 2020). Recently, a novel epithelial cell population that is found bridging the secretory and ciliated clusters (referred to as inflammatory epithelial transit cells) was found to yield the highest viral load in airway epithelial cells (Yoshida *et al.* 2021).

Once infected, the airways have been shown to become inflamed and damaged. Patients with severe COVID-19 have demonstrated acute diffuse alveolar damage and diffuse inflammatory infiltrates (consisting of interstitial and peribronchial lymphocytes and intra-alveolar macrophages) (Hou *et al.* 2020; Schaefer *et al.* 2020). Compared to moderate cases and controls, patients with critical COVID-19 disease exhibited epithelial cells with significant expression of chemokine-ligand encoding genes that promote recruitment of neutrophils, T cells and mast cells (Chua *et al.* 2020; Lee *et al.* 2020*c*). Severe cases also show greater enrichment for neutrophils, comprising >60% of cellular composition of upper airway samples (Chua *et al.* 2020). The recruitment of immune cells to sites of epithelial infection is an important early innate defence mechanism, regulated by the secretion of cytokines and chemokines by infected epithelial cells, which can help control inflammation and promote pathogen clearance.

To study the disease mechanisms and evaluate the efficacy of antiviral drugs, it is important that models of COVID-19 reproduce the physiology and pathology of

infection. Here, we briefly review the human models that are currently being used in COVID-19 research and drug development.

## Models of the airway

**Human infection challenge model.** The human infection challenge model has, for many decades, helped to shorten the timeline of new therapeutic drugs and vaccines, and to understand the role of viruses in disease pathogenesis. Edward Jenner's 1796 iconic challenge experiment demonstrated the protective effects of cowpox against smallpox and launched the field of vaccinology. We now have well-defined ethical criteria for human-challenge trials (WHO, 2020) and state-of-the-art facilities that satisfy safety and regulatory requirements, making this a scientifically acceptable and ethically valid method in the modern age. (For a review of ethical considerations, see Miller & Grady, 2001). Previous human infection challenge trials, of several hundred adult subjects, have established this as a safe and effective method in which to study the viral life cycle of many respiratory viruses including influenza virus, respiratory syncytial virus (RSV) and rhinovirus, the latter of which has been shown to reproduce the natural acquired infection (Lambkin-Williams *et al.* 2018). A human COVID-19 infection challenge trial is currently taking place at the Royal Free Hospital, London, infecting 90 healthy adults (aged 18–30) to determine the lowest possible dose of SARS-CoV-2 to cause disease (Callaway, 2020).

**Human *in vitro* models.** *In vitro* models are essential for the pre-clinical evaluation of potential therapeutics and can give new insight into the mechanism of infection and viral pathogenesis. Indeed, high-throughput cell line assays measuring cytopathic effects were instrumental in the discovery and development of anti-viral drugs including remdesivir (Eastman *et al.* 2020). Since the detection of SARS-CoV-2 in November 2019, several *in vitro* models have been used to determine replication, infection and cytopathic effects of the virus. The essential characteristics of cell models required for SARS-CoV-2 infection are thought to be the expression of ACE2 and, secondarily, spike priming proteases TMPRSS2 and furin. Investigations into the susceptibility of cell lines to SARS-COV-2 infections showed that replication was most robust in Calu-3 (a human lung adenocarcinoma cell line) and Caco2 (a human intestinal epithelial cell line), whilst hepatic Huh7, renal HEK293T and neuronal U251 cells were also able to propagate the virus (Chu *et al.* 2020). However, the commonly used lung adenocarcinoma cell line A549 (often used to model alveolar type II cells) and the immortalised human bronchial epithelium cell line BEAS2B are poor models for SARS-CoV-2 investigation

as they are not permissive to infection, unless transformed with ACE2 (Kam *et al.* 2009; Blanco-Melo *et al.* 2020; Chu *et al.* 2020). This is especially intriguing as their respective areas of origin (alveoli and bronchi) have demonstrated ACE2 expression and infection *in vivo* (Ziegler *et al.* 2020; Hou *et al.* 2020; Salahudeen *et al.* 2020). Other non-human primate cell lines that have been historically employed for virology and replicate SARS-CoV-2 include kidney cells FRhK4, LLCMK2 and Vero E6 cells (Matsuyama *et al.* 2020; Chu *et al.* 2020). Interestingly, the heavily used Vero E6 cell line does not express TMPRSS2 yet produces robust propagation of SARS-CoV-2 and is commonly used to generate viral stocks from clinical isolates (Harcourt *et al.* 2020; Ogando *et al.* 2020). A biochemical cleavage assay that uses Caco2, HEK293T or Vero E6 cells has been used to show SARS-CoV-2 spike protein harbours a distinct four-amino-acid insertion at the S1/S2 that can be cleaved by furin-like, trypsin-like and cathepsin proteases (Hoffmann *et al.* 2020; Jaimes *et al.* 2020). Whilst this is of interest for potential development of protease inhibitors (i.e. camostat) for therapeutics, a caveat here is the cathepsin spike priming pathway may only target the residual protein unprimed by TMPRSS2 and therefore act redundantly within COVID-19 disease severity and progression (Hoffmann *et al.* 2020).

It is also crucial to note that whilst passaging SARS-CoV-2 in cell lines, and specifically Vero E6 cells, the virus is under strong selection pressure to acquire adaptive mutations in its spike protein gene, due to the lack of relevant protease expression (Klimstra *et al.* 2020; Ogando *et al.* 2020; Peacock *et al.* 2020). This selection pressure can lead to attenuated replication in human bronchial epithelial cells (Pohl *et al.* 2021). These mutations may be negated by propagating SARS-CoV-2 in the serine protease expressing Calu-3 cell line (Lamers *et al.* 2021). These are important considerations for drug discovery. For example, a screen of ∼3000 potential anti-viral drugs using Vero E6 cells showed few hits, but the same library screened in Calu-3 revealed nine hits, seven of which are currently being used in human trials (Dittmar *et al.* 2021). Development of SARS-CoV-2 pseudoviruses also offers a potential solution for the genetic instability apparent with Vero E6 propagation, with the additional benefit of increased safety, reproducibility and scalability for screening assays (for reviews see Li *et al.* 2018; Chen & Zhang, 2021). Proof of principle for this has been demonstrated with a HIV-based lentiviral pseudovirus assay in a HEK293T cell line expressing human ACE2 and TMPRSS2 (Neerukonda *et al.* 2021). This assay has shown success in screening for neutralizing antibodies for SARS-CoV-2 (Neerukonda *et al.* 2021).

Calu-3 cells have also been a useful model in determining the function of SARS-CoV-2 non-structural proteins (NSPs). Employing a range of functional analyses

including RNA/protein crosslinking, splice reporter assays and surface sensing of translation (SUnSET) assays, NSPs were shown to disrupt mRNA processing. Mechanisms included suppression of global mRNA splicing (NSP16), binding to 18S ribosomal RNA causing global inhibition of mRNA translation (NSP1) and disruption of protein trafficking to the cell membrane (NSP8 and NSP9) (Banerjee *et al.* 2020).

Whilst easy to obtain, with the generation of reproducible assays, these immortalised cell cultures (using conventional submerged culture techniques) lack appropriate cell polarisation and many other distinguishing properties found in the lung, such as transport proteins, mucus and motile cilia. Fortunately, the relative accessibility to airway epithelium and air–liquid interface (ALI) culture techniques (described below) has allowed the development of a gold standard for *in vitro* epithelial models using primary airway epithelial cells.

**Air–liquid interface cell culture.** The ALI culture platform allows a physiological and pragmatic *in vitro* model that resides at the top of the experimental hierarchy for pre-clinical airway epithelial model systems (Karp *et al.* 2002). Here, cells are grown on semipermeable membranes with a basolateral medium supply and apical exposure to air in a well humidified (>95%) environment. This method is commonly used in a 24- or 12-well plate format, but recently groups have demonstrated the application of a miniaturised 96-well microplate system (Hyang Lee *et al.* 2020) and robotics for exchange of media that enable higher throughput drug testing (Bluhmki *et al.* 2020). Exposure to air stimulates the epithelial progenitor cells to differentiate into heterogeneous pseudostratified ciliated, goblet and basal cells, alongside other specialised cell types such as the ionocyte (Montoro *et al.* 2018). After 28 days, this differentiated epithelium exhibits mucus production and motile cilia, and transcriptionally has been shown to strongly correlate (>96% similarity) with the expression profile of epithelial cells obtained from the original nasal brushings from the same patients (Ghosh *et al.* 2020).

There are two major advantages of the ALI system. First, it allows for several precise, functional readouts of airway physiology, including the measurement of ciliary beating, mucociliary clearance, current or voltage across the membrane, differential protein secretion, airway surface liquid height measurements, ion transport measurements and wound healing assays (see review articles: Gianotti *et al.* 2018; Hiemstra *et al.* 2019). Secondly, airway epithelial cell culture models can be generated from any donor of interest, allowing the modelling of a range of human phenotypes. For example, models can be developed from donors that are diverse for variables associated with COVID-19 severity, such as age, ethnicity, sex, comorbidities such as diabetes and smoking/vaping

and respiratory disorders. All of these have demonstrated distinct phenotypic and/or functional characteristics in ALI culture (Clunes *et al.* 2012; Bilodeau *et al.* 2016; Castellani *et al.* 2018; Carlier *et al.* 2018; Gaiha *et al.* 2020; Woodall *et al.* 2020; Zhu *et al.* 2021).

**Primary ALI cell cultures.** Primary airway progenitor cells are the principal choice for the ALI method. These cells can be refined from small tissue biopsies taken from donated lungs, nasal brushings or bronchial brushings by bronchoscopy. The main challenge of working with primary cell culture lies with procurement of the appropriate cells and access to ethically sourced lung/airway tissue. Even when readily available, the process of retrieving these tissues requires specialist and expensive culture methods and carries risk of contamination. There is also a substantial amount of evidence that shows that different culture practices, including different medium preparations, membrane pore size and collagen coating method, and the duration at ALI can vary the characteristics of the epithelial model (Gianotti *et al.* 2018; Lee *et al.* 2020*a*; Leung *et al.* 2020).

In traditional airway epithelial cell culture systems, primary cells do not survive for more than a few passages and their characteristics can deteriorate rapidly with age. One solution of many (see review article: Orr & Hynds, 2021) to increase proliferative capacity of the progenitor cells has been to expand the progenitor cells in co-culture with irradiated/mitotically inactivated fibroblasts prior to conversion to ALI for differentiation (Butler *et al.* 2016). Another solution has been to introduce anti-senescent mechanisms such as viral oncogenes or the human poly-comb protein BMI-1 (Fulcher *et al.* 2009; Munye *et al.* 2016; Gianotti *et al.* 2018). The use of induced pluripotent stem cells (iPSCs) is also becoming increasingly popular, as fully differentiated cells such as fibroblasts can be expanded rapidly as iPSCs and then reprogrammed to differentiate into club cells, goblet cells or ciliated cells that self-assemble into a functional pseudostratified airway epithelium (Hawkins *et al.* 2021), reproduce characteristics of the proximal and distal airways (Pollard & Pollard, 2018) and are capable of replicating SARS-CoV-2 (Huang *et al.* 2020*b*).

**Immortalised cell lines.** Some immortalised airway epithelial cell lines, such as 16HBE14, and spontaneous cancer cells (i.e. Calu-3 and H441) (see table in review article: Orr & Hynds, 2021) that are cultured on semi-permeable membranes can form tight junctions, secrete mucins, and show differential distribution of plasma membrane transport proteins between the apical membrane and the basolateral membrane epithelium (Castellani *et al.* 2018), but do not produce ciliated and other important specialised cell types. Calu-3 cells

are a popular epithelial cell line for infection studies as they are known to generate higher transepithelial resistance, produce mucus, and express a diverse set of immune and inflammatory modulators (Grainger et al. 2006; Braakhuis et al. 2020). The HBEC3 series are thought to be the most favourable airway model due to generation of motile cilia (Lodes et al. 2020). To facilitate the study of basal cell biology, a few research groups have successfully immortalised basal cell lines expanding the experimental scope. In one instance, basal cell immortalised non-smoker 1 (BCi-NS1) is an immortalised human large airway basal cell line generated via retrovirus-mediated expression of human telomerase (hTERT) (Walters et al. 2013). The small airway epithelium basal cell line hSABCi-NS1.1 was generated via a similar methodology (Wang et al. 2019). The development of these basal cell lines with a multipotent differentiation capacity retaining characteristics to the original primary basal cells for over 40 passages is a useful tool for understanding basal cell biology, pathogenesis and related diseases allowing long term experimentation. Notably, expression of genes encoding ACE2, TMPRSS2 and other essential factors necessary for SARS-CoV-2 infection has been detected in BCi-NS1.1 and hSABCi-NS1.1 cell lines, thus advocating for their suitability in COVID-19 research (Zhang et al. 2020).

Whilst immortalised cell lines are pragmatic for high throughput assays, especially those required for drug screening, a resounding problem in their use is the inability to represent age phenotypes when age-related severity is a distinguishing characteristic of COVID-19.

**Primary ALI cultures for SARS-CoV-2 research.** Primary cells grown using ALI culture have been shown to propagate SARS-CoV-2 effectively in epithelial cell types expressing ACE2 (Hou et al. 2020; Robinot et al. 2020; Zhu et al. 2020, 2021; Pohl et al. 2021). Interestingly, the number of ciliated cells in airway epithelial cell cultures did not correlate with susceptibility to infection (Hou et al. 2020). This may be due to the presence of secondary entry receptors (NRP1, BSG, TFRC), whose expression has shown some correlation with actively infected cells in scRNA-seq studies and cell line assays (Wang et al. 2020; Tang et al. 2020; Cantuti-Castelvetri et al. 2020; Daly et al. 2020; Yoshida et al. 2021), or changes to epithelial defence mechanisms, including apparent basal cell mobilization (Robinot et al. 2020). ALI cultures have also been shown to support long- and short-term modelling of SARS-CoV-2. Short-term studies are useful to investigate virus replication and cytopathic effects, and there is rapidly emerging data suggesting some prevalent variants of SARS-CoV-2 may have altered replication dynamics within human airway epithelial cells, whilst

some in vitro passaged isolates may develop mutations that severely decrease infectivity of human airway epithelial cells (Peacock et al. 2020; Liu et al. 2021; Pohl et al. 2021). Long-term modelling allows us to study the airway epithelium's ability to repair and regenerate (Hao et al. 2020).

Functional readouts from ALI-specific assays have shown a transient decrease in epithelial barrier function and disruption of tight junctions, though infectious viral particles almost exclusively remain at the apical side of cultures (Fig. 1) (Robinot et al. 2020; Hao et al. 2020; Zhu et al. 2020). SARS-CoV-2 infection also led to a rapid loss of the ciliary layer and this resulted in reduction of motile cilia function as measured in a mucociliary clearance assay (Robinot et al. 2020). Furthermore, iPSC-derived AT2 cultures demonstrated rapid transcriptomic change in SARS-CoV-2-infected cells to an inflammatory phenotype, characterised by an upregulation of nuclear factor $\kappa$B signalling (Huang et al. 2020b).

Studies have shown that ALI cultures display the age-, sex- and disease-dependent changes in ACE2 mRNA levels that have been observed in lung tissue from different donors (Lukassen et al. 2020). Specifically, the expression of putative SARS-CoV-2 receptors was found to be lower in the upper and lower airways in children compared to other age groups, whilst expression of both ACE2 and TMPRSS2 were upregulated in smokers and patients with chronic obstructive pulmonary disease compared with healthy subjects (Saheb Sharif-Askari et al. 2020). The role of entry factors in determining disease severity remains inconclusive, and other studies suggest that the immune response to SARS-CoV-2 may play a more important role (Koch et al. 2021), with elevated production of type I and III interferons, rather than differential receptor expression, suggested as the cause of the lower viral replication in paediatric compared to adult nasal epithelial cultures (Zhu et al. 2021).

**Trans-epithelial migration models.** The migration of neutrophils and other polymorphonuclear leukocytes (PMNs) across columnar epithelia is a key component of mucosal defence and inflammation, and it is becoming increasingly evident that COVID-19 disease severity is associated with a dysregulation of the immune response, including increased neutrophil recruitment (Lee et al. 2020b). To study this further, a modified ALI model has been developed that facilitates neutrophil transepithelial migration in response to airway infection or other stimuli (for review see Adams et al. 2021). This three-dimensional in vitro technique differs from the aforementioned ALI model in that airway epithelial cells are cultured on the underside of the membrane insert. The inverted insert is coated with collagen to assist in the epithelial cell

attachment. Once fully differentiated (at ALI for 28 days) and infected, neutrophils or PMNs are added to the baso-lateral (medium) side of the epithelial cells so they can migrate in the physiological direction, from basolateral to apical surface. This model has already revealed important insights into the contribution of neutrophils to airway damage and viral clearance during RSV infection (Deng *et al.* 2018; Herbert *et al.* 2020). Additionally, this model was used to study neutrophil transepithelial migration during *Pseudomonas aeruginosa* infection (Kusek *et al.* 2014; Yonker *et al.* 2017). Although this model has not yet been utilised for COVID-19 research, work is underway in our group to study neutrophil transepithelial migration in response to SARS-CoV-2 infection.

Immortalised cell lines have also been applied in co-culture models. Specifically, Calu-3 cells have been used as the structural barrier to create co-culture models with human macrophages, to model inflammatory responses to aerosols (Grainger *et al.* 2006).

**Organoids.** Another *in vitro* model that closely recapitulates lung epithelial function and architecture is three dimensional organoids. These self-assembling structures can be cultured in a three dimensional extracellular matrix using a variety of progenitor cells including basal cells, airway secretory club cells, epithelial cells, iPSC and crypt stem cells (Lancaster & Knoblich, 2014; Barkauskas *et al.* 2017). Though there are some limitations with these models, such as the restricted accessibility to the luminal surface, they are quickly becoming valuable in development of personalised therapies (Dekkers *et al.* 2016; Berkers *et al.* 2019), to study lung development (Vazquez-Armendariz et al. 2020) and infection with viruses such as rotavirus, norovirus, enterovirus 71 and human adenovirus (Ramani *et al.* 2018). Both airway and intestinal organoids have been shown to express high levels of ACE2 and membrane-bound serine proteases TMPRSS2 and TMPRSS4 enabling cleavage of the SARS-CoV-2 spike protein to facilitate viral entry (Zang *et al.* 2020; Suzuki *et al.* 2020). Indeed, three simultaneous studies (Zang *et al.* 2020; Zhou *et al.* 2020a; Lamers *et al.* 2020) used human adipose-derived stem-derived intestinal organoids to provide evidence that SARS-CoV-2 could establish itself in the gastrointestinal tract, showing that the most common cell type of the intestinal epithelium, the enterocyte, is readily infected and strongly upregulates viral response genes (Lamers *et al.* 2020). One recent development is the generation of distal lung organoids by embedding cells in extracellular matrices to form cyst-like organoids with apical-out polarity (Danahay *et al.* 2015; Lukassen *et al.* 2020) to present ACE2 on the exposed external surface (Salahudeen *et al.* 2020). This polarity allows for the more physiological, non-invasive apical infection of AT2 and basal cultures with SARS-CoV-2, already leading to the identification of club cells as another target population via scRNA-seq analysis (Salahudeen *et al.* 2020).

Organoid cultures have also shown potential for studying immune cell–epithelium interactions. The co-culture of human intestinal stem cell-derived enteroid monolayers with human monocyte-derived macrophages have demonstrated communication between the epithelium and macrophages through morphological changes and cytokine production in response to *E. coli* infections (Noel *et al.* 2017).

**Lung-on-a-chip.** A precursor to the lung organoid culture was the 'lung-on-a-chip' model. This biological device uses microfluidics technologies and allows

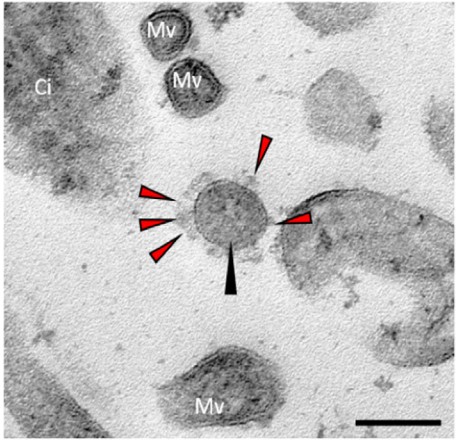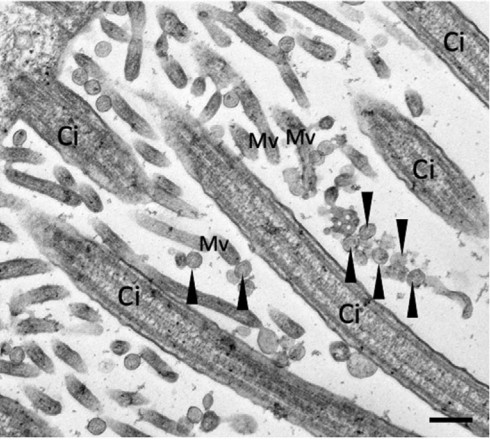

**Figure 1. Transmission electron micrographs of SARS-CoV-2-infected ciliated nasal epithelial cells grown in culture at an air–liquid interface**
Black arrows indicate SARS-CoV-2 viral particles; red arrowheads indicate the viral spike protein. Ci, cilia; Mv, microvilli. Scale bar: 200 nm. Image credit Andreia Pinto.

communication between epithelial and endothelial cells through a porous and elastic membrane. Here, epithelial cells in the upper chamber are exposed to the ALI, whilst the submerged endothelial cells (grown in the bottom layer) co-ordinate the uptake of nutrients from the media (Benam *et al.* 2016*b*). The lung-on-a-chip model has been employed successfully in some drug discovery and toxicity studies (Huh *et al.* 2012; Gkatzis *et al.* 2018), including chips connected to an instrument that mimics 'breathing' (Benam *et al.* 2016*a*; Huang *et al.* 2021). Recently, the human airway-on-a-chip model has shown utility in testing potential antiviral therapeutics against COVID-19 using pseudotyped (Si *et al.* 2021) and wild-type SARS-CoV-2 virus (Thacker *et al.* 2021; Deinhardt-Emmer *et al.* 2021).

### Future directions

**Ion transport in COVID-19.** Despite the consensus that airway epithelial cell function is key in COVID-19 pathogenesis, there has been a lack of research on how critical epithelial functions, such as ion transport, are affected by SARS-CoV-2 infection (Gentzsch & Rossier, 2020). For example, it has been suggested that SARS-CoV-2 could disrupt conserved second messenger signalling cascades via G protein-coupled receptors, adversely modulating transepithelial transport processes (Hameid *et al.* 2021). The role of cystic fibrosis transmembrane conductance regulator (CFTR) in COVID-19 is also undescribed and may be of significance as a key regulator of mucociliary clearance and airway liquid pH, especially since SARS-CoV-2 entry into airway epithelial cells is pH-dependent (Hoffmann *et al.* 2020; Shang *et al.* 2020). Investigation here may be valuable as it has been shown in an influenza model that potentiating CFTR expression and function with corrector lumacaftor reverses *in vitro* down-regulation of CFTR and ENaC following viral infection, rehydrating the airway surface liquid (Brand *et al.* 2018), which could aid viral clearance.

**Tissue engineering.** Advances in whole lung bioengineering using engineered three-dimensional scaffolds and microenvironments have opened new possibilities for studying lung regeneration and infection *ex vivo* using acellular human and non-human derived lung tissue scaffolds. Methods to decellularize whole human lungs, lobes or resected segments from normal and diseased human lungs have been developed using both perfusion and immersion-based techniques (Lin *et al.* 2009; Asnaghi *et al.* 2009; Castellani *et al.* 2018). These bioreactors, containing for example autologous respiratory epithelial cells and mesenchymal stem cells (BMSCs, then differentiated into chondrocytes), have been used clinically in tracheal transplantation (Macchiarini *et al.* 2008; Day, 2019) but also may be able to predict oxygen profiles (Asnaghi *et al.* 2009) following infection. These cellular systems, combined with improved sensitivity in readouts, offer immense potential to study the functional responses to respiratory virus infection using superior, physiologically relevant human models.

### Conclusion

Every experimental model of the human airway has its limitations. Whilst cell lines are pragmatic models for reproducible high throughput assays, complex primary cell models, produced from donors from a range of demographics, are the only ones capable of representing the variability of disease severity that has become characteristic of COVID-19 disease.

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

## Additional information

### Competing interests

All authors declare no competing interests.

### Author contributions

M.W. contributed to paper conception, manuscript preparation and data collection (Fig. 1). T.M. and K.C. contributed to manuscript preparation. C.M.S. contributed to paper conception, manuscript preparation and graphical abstract. All authors have read and approved the final version of this manuscript and agree to be accountable for all aspects of the

work in ensuring that questions related to the accuracy or integrity of any part of the work are appropriately investigated and resolved. All persons designated as authors qualify for authorship, and all those who qualify for authorship are listed.

## Funding

This work was supported by the National Institute for Health Research (NIHR) Great Ormond Street Hospital Biomedical Research Centre. The views expressed are those of the authors and not necessarily those of the NHS, the NIHR or the Department of Health. T.M. would also like to thank A.G Leventis Foundation for their support and help during her studies. C.M.S is currently a recipient of grants from Animal Free Research UK (AFR19-20274), BBSRC (BB/V006738/1), GOSH Children's charity (COVID_CSmith_017) and the Wellcome Trust (212516/Z/18/Z).

Funding for Open Access was approved for this article.

## Acknowledgements

All authors want to thank Dr Robert Hynds (University College London and The Francis Crick Institute, London) for feedback on draft manuscripts, and Dr Andreia Pinto (Royal Brompton Hospital, London) for providing transmission electron microscopy images.

## Keywords

cilia, differentitated, infection, respiratory, SARS-CoV-2

## Supporting information

Additional supporting information can be found online in the Supporting Information section at the end of the HTML view of the article. Supporting information files available:

**Peer Review History**
**Video abstract and summary**

