## [Peer Review History · The Journal of Physiology]

TBD - from "COVID-19 Lessons Learned from the Frontline" Webinar

Maximillian Woodall, Katie Marie Case, Tereza Masonou, and Claire M. Smith
DOI: 10.1113/JP281499

Corresponding author(s): Claire Smith (c.m.smith@ucl.ac.uk)

Review Timeline:	Submission Date:	23-Apr-2021
	Editorial Decision:	24-May-2021
	Revision Received:	18-Jun-2021
	Accepted:	12-Jul-2021

Senior Editor: Ian Forsythe

Reviewing Editor: Frank Powell

Transaction Report:

Dear Dr Smith,

Re: JP-TR-2021-281499 "TBD - from "COVID-19 Lessons Learned from the Frontline" Webinar" by Maximillian Woodall, Katie Marie Case, Tereza Masonou, and Claire M. Smith

Thank you for submitting your Topical Review to The Journal of Physiology. It has been assessed by a Reviewing Editor and by 2 expert referees and I pleased to tell you that it is considered to be acceptable for publication following satisfactory revision.

The reports are copied at the end of this email. Please address all of the points and incorporate all requested revisions, or explain in your Response to Referees why a change has not been made.

NEW POLICY: In order to improve the transparency of its peer review process The Journal of Physiology publishes online as supporting information the peer review history of all articles accepted for publication. Readers will have access to decision letters, including all Editors' comments and referee reports, for each version of the manuscript and any author responses to peer review comments. Referees can decide whether or not they wish to be named on the peer review history document.

I hope you will find the comments helpful and have no difficulty in revising your manuscript within 4 weeks.

Your revised manuscript should be submitted online using the links in Author Tasks Link Not Available. This link is to the Corresponding Author's own account, if this will cause any problems when submitting the revised version please contact us.

You should upload:

- A Word file of the complete text (including any Tables);
- An Abstract Figure, (with accompanying Legend in the article file)
- Each figure as a separate, high quality, file;
- A full Response to Referees;
- A copy of the manuscript with the changes highlighted.
- Author profile. A short biography (no more than 100 words for one author or 150 words in total for two authors) and a portrait photograph of the two leading authors on the paper. These should be uploaded, clearly labelled, with the manuscript submission. Any standard image format for the photograph is acceptable, but the resolution should be at least 300 dpi and preferably more.

- A 'Cover Art' file for consideration as the Issue's cover image;
- Appropriate Supporting Information (Video, audio or data set https://jp.msubmit.net/cgi-bin/main.plex?form_type=display_requirements#supp).

To create your 'Response to Referees' copy all the reports, including any comments from the Senior and Reviewing Editors into a Word, or similar, file and respond to each point in colour or CAPITALS. Upload this when you submit your revision.

I look forward to receiving your revised submission.

Yours sincerely,

Ian D. Forsythe
Deputy Editor-in-Chief
The Journal of Physiology
<https://jp.msubmit.net>
<http://jp.physoc.org>
The Physiological Society
Hodgkin Huxley House
30 Farringdon Lane
London, EC1R 3AW
UK
<http://www.physoc.org>
<http://journals.physoc.org>

EDITOR COMMENTS

Reviewing Editor:

Your work has been reviewed by two experts who both find considerable merit with the review but they also have suggestions for minor changes and improvements for consideration for a final decision. Also, please note that Ref. 1 noted that the work could benefit from clearly stating if the referenced literature used nasal or bronchial epithelial cells.

REFEREE COMMENTS

Referee #1:

This is a nice overview of the cell culture methods used in respiratory research that have applicability for COVID-19 research. I have the following comments:

- 1.) Please be more specific about the origin of the cells (nasal, bronchial, small bronchiole, alveolar) when you give examples.
- 2.) paragraph 'Primary ALI cell cultures': 'These cells can be refined from small tissue biopsies...or nasal brushings'. I recommend to include bronchial brushings by bronchoscopy
- 3.) paragraph immortalised cell lines: I would recommend to include hTERT transformed primary cells here as well (e.g. BCI cell line)
- 4.) for consistency, please check that it is ACE2, in some places it is ACE-2
- 5.) paragraph 'organoids', second paragraph has missing words. monocyte-derived 'macrophages'?
- 6.) the section of lung-on-a-chip can be much improved. The model described here is an 'alveoli'

on a chip. There are more microfluidic models out there that use e.g. primary differentiated bronchial epithelial cells which are worth mentioning. Also worth mentioning that chip based models have the potential for automated online monitoring, which has great potential for work in Cat3 labs

Referee #2:

The authors have done a scholarly review of the existing literature regarding various models of sars-covll. The article is well written and provides a useful synthesis of a large amount of data. This reviewer has little to add although a few additions may be useful e.g. work by Clausen in Cell using organoids and other approaches. Also efforts to use the spike protein or pseudo-spike to study endothelial cells have provided important insights. a brief review of how the various models have informed other viral illnesses may also provide some important context

REQUIRED ITEMS:

-Your MS must include a complete "Additional information section" with the following 4 headings and content:

Competing Interests: A statement regarding competing interests. If there are no competing interests, a statement to this effect must be included. All authors should disclose any conflict of interest in accordance with journal policy.

Author contributions: Each author should take responsibility for a particular section of the study and have contributed to writing the paper. Acquisition of funding, administrative support or the collection of data alone does not justify authorship; these contributions to the study should be listed in the Acknowledgements. Additional information such as 'X and Y have contributed equally to this work' may be added as a footnote on the title page.

It must be stated that all authors approved the final version of the manuscript and that all persons designated as authors qualify for authorship, and all those who qualify for authorship are listed.

Funding: Authors must indicate all sources of funding, including grant numbers. If authors have not received funding, this must be stated.

It is the responsibility of authors funded by RCUK to adhere to their policy regarding funding sources and underlying research material. The policy requires funding information to be included within the acknowledgement section of a paper. Guidance on how to acknowledge funding information is provided by the Research Information Network. The policy also requires all research papers, if applicable, to include a statement on how any underlying research materials, such as data, samples or models, can be accessed. However, the policy does not require that

the data must be made open. If there are considered to be good or compelling reasons to protect access to the data, for example commercial confidentiality or legitimate sensitivities around data derived from potentially identifiable human participants, these should be included in the statement.

Acknowledgements: Acknowledgements should be the minimum consistent with courtesy. The wording of acknowledgements of scientific assistance or advice must have been seen and approved by the persons concerned. This section should not include details of funding.

-Author profile(s) must be uploaded via the submission form. Authors should submit a short biography (no more than 100 words for one author or 150 words in total for two authors) and a portrait photograph of the two leading authors on the paper. These should be uploaded, clearly labelled, with the manuscript submission. Any standard image format for the photograph is acceptable, but the resolution should be at least 300 dpi and preferably more. A group photograph of all authors is also acceptable, providing the biography for the whole group does not exceed 150 words.

-It is the authors' responsibility to obtain any necessary permissions to reproduce previously published material https://jp.msubmit.net/cgi-bin/main.plex?form_type=display_requirements#use

END OF COMMENTS

Confidential Review

23-Apr-2021

Response to referees comments (in bold)

REFEREE COMMENTS

Referee #1:

1.) Please be more specific about the origin of the cells (nasal, bronchial, small bronchiole, alveolar) when you give examples. Any examples of airway cells that have multiple references have not been changed in order to not disrupt the flow of the text. Single references have now been changed to the specific origin of the cells.

2.) paragraph 'Primary ALI cell cultures': 'These cells can be refined from small tissue biopsies...or nasal brushings'. I recommend to include bronchial brushings by bronchoscopy. This was added.

3.) paragraph immortalised cell lines: I would recommend to include hTERT transformed primary cells here as well (e.g. BCI cell line) Basal cell biology has been addressed within the paragraph of immortalised cell lines referencing Walters et al 2013, Wang et al 2019 and Zhang et al 2020.

4.) for consistency, please check that it is ACE2, in some places it is ACE-2 Consistency was checked for ACE2, COVID-19 and SARS-CoV-2 throughout the text, any inconsistencies have now been changed.

5.) paragraph 'organoids', second paragraph has missing words. monocyte-derived 'macrophages'? This was added.

6.) The section of lung-on-a-chip can be much improved. The model described here is an 'alveoli' on a chip. There are more microfluidic models out there that use e.g. primary differentiated bronchial epithelial cells which are worth mentioning. Also worth mentioning that chip based models have the potential for automated online monitoring, which has great potential for work in Cat3 labs . We agree that this section could use more detail and have changed the descriptions of our lung on chip to refer to other systems outside the lung and added further references to other systems and including an instrument that mimics “breathing” (Benam et al., 2016a; Huang et al., 2021). We also added description of very recent work using a human bronchial airway-on-a-chip model that has shown utility in testing potential antiviral therapeutics against COVID-19 (Si et al., 2021).

Referee #2:

The authors have done a scholarly review of the existing literature regarding various models of sars-covII. The article is well written and provides a useful synthesis of a large amount of data. This reviewer has little to add although a few additions may be useful e.g. work by Clausen in Cell using organoids and other approaches. We thank the reviewer for their comments and suggestions, however we think that this particular reference maybe outside the remit of the subject area of this review. This paper describes cardiac organoids, whereas we would like to keep our review focused on airway organoids.

(<https://www.sciencedirect.com/science/article/pii/S1934590919301080?via%3Dihub>)

Also efforts to use the spike protein or pseudo-spike to study endothelial cells have provided important insights. a brief review of how the various models have informed other viral illnesses may also provide some important context. We have added a section discussing proof of principle of this in a HIV-based lentiviral pseudovirus assay:

<https://journals.plos.org/plosone/article?id=10.1371/journal.pone.0248348>. We thank the reviewer for their comments and suggestions, however we think that we have given some relevant examples of this where appropriate but do not wish to distract from keeping our review focused SARS-CoV-2.

Additional information section with the following 4 headings and content:

1.) Competing Interests: A statement regarding competing interests. If there are no competing interests, a statement to this effect must be included. All authors should disclose any conflict of interest in accordance with journal policy. We have added this section.

2.) Author contributions: Each author should take responsibility for a particular section of the study and have contributed to writing the paper. Acquisition of funding, administrative support or the collection of data alone does not justify authorship; these contributions to the study should be listed in the Acknowledgements. Additional information such as 'X and Y have contributed equally to this work' may be added as a footnote on the title page. We have added this section.

3.) It must be stated that all authors approved the final version of the manuscript and that all persons designated as authors qualify for authorship, and all those who qualify for authorship are listed. We have added this section.

4.) Funding: Authors must indicate all sources of funding, including grant numbers. If authors have not received funding, this must be stated. We have added this section.

It is the responsibility of authors funded by RCUK to adhere to their policy regarding funding sources and underlying research material. The policy requires funding information to be included within the acknowledgement section of a paper. Guidance on how to acknowledge funding information is provided by the Research Information Network. The policy also requires all research papers, if applicable, to include a statement on how any underlying research materials, such as data, samples or models, can be accessed. However, the policy does not require that the data must be made open. If there are considered to be good or compelling reasons to protect access to the data, for example commercial confidentiality or legitimate sensitivities around data derived from potentially identifiable human participants, these should be included in the statement. **done**

Acknowledgements: Acknowledgements should be the minimum consistent with courtesy. The wording of acknowledgements of scientific assistance or advice must have been seen and approved by the persons concerned. This section should not include details of funding. We have added this section

-Author profile(s) must be uploaded via the submission form. Authors should submit a short biography (no more than 100 words for one author or 150 words in total for two authors) and a portrait photograph of the two leading authors on the paper. These should be uploaded, clearly labelled, with the manuscript submission. Any standard image format for the photograph is acceptable, but the resolution should be at least 300 dpi and preferably more. A group photograph

of all authors is also acceptable, providing the biography for the whole group does not exceed 150 words. See Below

Dr. Claire M. Smith, PhD, is a Lecturer in the department of Infection, Immunity and Inflammation, GOS Institute of Child Health, University College London. Her research focuses on anti-viral therapies, mucosal immunity and the interaction between the airway epithelium and neutrophils during respiratory virus infection.

Dr. Maximillian Woodall, PhD, is a Research Fellow working alongside Dr. CMS. His research employs an experimental model of the human airway to investigate age-dependent factors driving COVID-19 disease severity.

Miss Tereza Masonou, MSc, is a PhD student working under the supervision of Dr. CMS. Her research investigates neutrophil trans-epithelial migration and function during SARS-CoV2 infection, and how this differs between older adults and children.

Miss Katie M. Case, MSc, is a Research Assistant working alongside Dr. MW. She maintains airway epithelial cells from different age groups and explores baseline differences such as cell motility, cilia beating and protein levels.

Dear Dr Smith,

Re: JP-TR-2021-281499R1 "TBD - from "COVID-19 Lessons Learned from the Frontline" Webinar" by Maximillian Woodall, Katie Marie Case, Tereza Masonou, and Claire M. Smith

I am pleased to tell you that your Topical Review article has been accepted for publication in The Journal of Physiology, subject to any modifications to the text that may be required by the Journal Office to conform to House rules.

NEW POLICY: In order to improve the transparency of its peer review process The Journal of Physiology publishes online as supporting information the peer review history of all articles accepted for publication. Readers will have access to decision letters, including all Editors' comments and referee reports, for each version of the manuscript and any author responses to peer review comments. Referees can decide whether or not they wish to be named on the peer review history document.

The last Word version of the paper submitted will be used by the Production Editors to prepare your proof. When this is ready you will receive an email containing a link to Wiley's Online Proofing System. The proof should be checked and corrected as quickly as possible.

All queries at proof stage should be sent to tjp@wiley.com

The accepted version of the manuscript will be published online, prior to copy editing in the Accepted Articles section.

Are you on Twitter? Once your paper is online, why not share your achievement with your followers. Please tag The Journal (@jphysiol) in any tweets and we will share your accepted paper with our 22,000+ followers!

Yours sincerely,

Ian D. Forsythe
Deputy Editor-in-Chief
The Journal of Physiology
<https://jpp.msubmit.net>
<http://jpp.physoc.org>
The Physiological Society
Hodgkin Huxley House
30 Farringdon Lane
London, EC1R 3AW
UK
<http://www.physoc.org>
<http://journals.physoc.org>

*** IMPORTANT NOTICE ABOUT OPEN ACCESS ***

Information about Open Access policies can be found here
<https://physoc.onlinelibrary.wiley.com/hub/access-policies>

To assist authors whose funding agencies mandate public access to published research findings sooner than 12 months after publication The Journal of Physiology allows authors to pay an

open access (OA) fee to have their papers made freely available immediately on publication.

You will receive an email from Wiley with details on how to register or log-in to Wiley Authors Services where you will be able to place an OnlineOpen order.

You can check if your funder or institution has a Wiley Open Access Account here <https://authorservices.wiley.com/author-resources/Journal-Authors/licensing-and-open-access/open-access/author-compliance-tool.html>

Your article will be made Open Access upon publication, or as soon as payment is received.

If you wish to put your paper on an OA website such as PMC or UKPMC or your institutional repository within 12 months of publication you must pay the open access fee, which covers the cost of publication.

OnlineOpen articles are deposited in PubMed Central (PMC) and PMC mirror sites. Authors of OnlineOpen articles are permitted to post the final, published PDF of their article on a website, institutional repository, or other free public server, immediately on publication.

Note to NIH-funded authors: The Journal of Physiology is published on PMC 12 months after publication, NIH-funded authors DO NOT NEED to pay to publish and DO NOT NEED to post their accepted papers on PMC.

EDITOR COMMENTS

Reviewing Editor:

Thank you for this responsive revision on your interesting scholarly review of this timely topic.

REFEREE COMMENTS

Referee #1:

The authors have addressed my points adequately and improved the manuscripts. No further comments from my side.

Referee #2:

No further comments.

1st Confidential Review

18-Jun-2021